# The Contemporary Role of Hematopoietic Stem Cell Transplantation in the Management of Chronic Myeloid Leukemia: Is It the Same in All Settings?

**DOI:** 10.3390/cancers16040754

**Published:** 2024-02-12

**Authors:** Ibrahim Elmakaty, Giuseppe Saglio, Murtadha Al-Khabori, Abdelrahman Elsayed, Basant Elsayed, Mohamed Elmarasi, Ahmed Adel Elsabagh, Awni Alshurafa, Elrazi Ali, Mohamed Yassin

**Affiliations:** 1College of Medicine, QU Health, Qatar University, Doha P.O. Box 2713, Qatar; 2Department of Clinical and Biological Sciences, University of Turin, 10124 Turin, Italy; 3Hematology Department, Sultan Qaboos University, Muscat 123, Oman; 4Hematology Section, Medical Oncology, National Center for Cancer Care and Research (NCCCR), Hamad Medical Corporation (HMC), Doha P.O. Box 3050, Qatar; 5Interfaith Medical Center, Brooklyn, NY 11213, USA

**Keywords:** chronic myeloid leukemia, hematopoietic stem cell transplant, tyrosine kinase inhibitors, treatment-free remission, BCR–ABL1 gene fusion, survival

## Abstract

**Simple Summary:**

This review explores the use of hematopoietic stem cell transplantation (HSCT) as a treatment option for chronic myeloid leukemia (CML) patients. While CML treatment has greatly improved with tyrosine kinase inhibitors (TKIs), some patients do not respond well or reach advanced disease stages. The goal now is to achieve treatment-free remission (TFR). Although discontinuing TKIs shows promise, relapse risk is high. This review discusses recent advances in HSCT and its role in CML treatment, suggesting it should be considered early in disease management to enhance the chances of achieving TFR alongside TKIs.

**Abstract:**

Hematopoietic stem cell transplantation (HSCT) for chronic myeloid leukemia (CML) patients has transitioned from the standard of care to a treatment option limited to those with unsatisfactory tyrosine kinase inhibitor (TKI) responses and advanced disease stages. In recent years, the threshold for undergoing HSCT has increased. Most CML patients now have life expectancies comparable to the general population, and therefore, the goal of therapy is shifting toward achieving treatment-free remission (TFR). While TKI discontinuation trials in CML show potential for achieving TFR, relapse risk is high, affirming allogeneic HSCT as the sole curative treatment. HSCT should be incorporated into treatment algorithms from the time of diagnosis and, in some patients, evaluated as soon as possible. In this review, we will look at some of the recent advances in HSCT, as well as its indication in the era of aiming for TFR in the presence of TKIs in CML.

## 1. Introduction

Chronic myeloid leukemia (CML) is a clonal hematopoietic stem cell illness defined by oncogenic breakpoint cluster region–Abelson (BCR–ABL1) gene fusion [1]. It is distinguished by the Philadelphia chromosome, which results from a reciprocal translocation of chromosomes 9 and 22 [2]. This chromosomal abnormality places the ABL1 gene next to the breakpoint cluster region gene, causing the fused BCR–ABL1 oncogene [3]. This dysregulated BCR–ABL1 protein phosphorylates multiple substrate proteins, resulting in a loss of cell-cycle regulation and a consequent increase in proliferation, loss of stromal adherence, and resistance to apoptosis [4].

Approximately 5–10% of CML patients lack the Philadelphia chromosome but exhibit detectable BCR–ABL1 oncogenes, with 25–50% displaying BCR–ABL1 gene rearrangement outside the Philadelphia chromosome [5]. The 2022 update from the World Health Organization (WHO) brought a notable change regarding BCR–ABL-negative CML, previously referred to as atypical CML. This condition has been reclassified as myelodysplastic/myeloproliferative (MDS/MPN) with neutrophilia [6]. This alteration highlights that the disease closely resembles MDS/MPN and aims to prevent any misunderstandings with traditional CML [6]. MDS/MPN with neutrophilia patients have a worse overall survival (OS) and are more likely to develop acute myeloid leukemia [7].

The chronic, accelerated, and blast phases are the three stages of CML [8]. Notably, the 2022 WHO edition eliminated the accelerated phase [6], categorizing CML patients into chronic and blast phases, although this change awaits endorsement from other guidelines [9]. This proposed classification shift likely stems from the fact that CML cases with less than 20% blasts exhibit favorable survival rates and a low progression incidence. Thus, merging the accelerated phase with the chronic phase streamlines treatment strategies.

With the emergence of targeted medicines such as imatinib mesylate and other tyrosine kinase inhibitors (TKIs), CML has seen considerable improvements in life expectancy and reduced life years lost [10,11,12]. The transformational effect of these medicines is visible in the reduction in age-standardized death rates, with yearly percent changes ranging from −11.6% to −20.8%, resulting in a mortality rate in 2008 that was approximately 30% lower than that in 1993 [11]. The death rate in CML patients has fallen by 50–80%, resulting in a significant rise in five-year relative survival across all age categories, notably in older patients [12]. The most common cause of mortality in CML is disease progression, stressing the significance of appropriate treatment despite the dangers of medication-induced adverse effects [13].

Before the development of medicinal therapy, the gold standard treatment for CML was allogeneic hematopoietic stem cell transplantation (HSCT) within one year of diagnosis [14]. For many years, HSCT has been used to treat a variety of hematologic disorders [15]. However, with the introduction of TKIs and their efficacy in CML treatment, HSCT has fallen out of favor due to treatment-related toxicities. While the difficulty in finding human leukocyte antigen (HLA)-matched donors has historically been a concern [16], it is noteworthy that the emergence of HLA–haploidentical bone marrow transplantation has provided a viable alternative for these patients, with outcomes comparable to those of matched donors [17].

Allogeneic HSCT is now considered the last-line therapeutic option for CML patients who have not responded to TKIs. According to a Center for International Blood and Marrow Transplant Research (CIBMTR) study, transplants peaked in 1999, when approximately 2000 allogeneic HSCTs were conducted in CML patients [18].

Post-HSCT complications encompass both acute and chronic issues. Acute complications, such as myelosuppression, mucositis, and acute graft-versus-host disease (GvHD), can arise in the early stages, while chronic complications, including chronic GvHD and infections, may persist over the long term [19]. Despite the similarity in survival rates between TKIs and HSCT, it is crucial to acknowledge that some HSCT patients may face persistent health challenges, notably due to chronic GvHD [19,20]. This condition can significantly impact their daily lives and overall well-being, influencing their long-term outcomes. The effective management of these complications is pivotal for enhancing the quality of life for HSCT recipients.

Treatment-free remission (TFR) has emerged as a new goal in managing CML, addressing concerns related to long-term toxicity, adverse side effects, and the financial burden of lifelong TKI therapy [20]. Achieving a deep molecular response (DMR) has been identified as a key factor for successful TFR, allowing patients to discontinue TKI treatment without experiencing disease relapse [21]. However, not all patients can achieve TFR, highlighting the importance of further investigation in this patient population.

Despite TKI dominance in CML therapy, new developments in allogeneic HSCT have improved patient outcomes and survival rates [22]. Although transplant-related mortality has been greatly reduced in recent decades due to significant improvements in transplant procedures and therapies used to control acute and chronic GvHD, the role of HSCTs in the treatment of CML is now considered marginal and reserved for specific cases in advanced stages of the disease [23].

Specific transplant recommendations for CML have been published by organizations such as European LeukemiaNet (ELN) [24], but it remains unclear whether these recommendations are universal or should be tailored to the economic and cultural circumstances of different parts of the world. The purpose of this review is to revisit the current indications for HSCT in the era of TKIs and TFR, as well as the factors that contribute to regional variation in its use.

## 2. Advancements and Considerations in HSCT for CML

### 2.1. Risk Stratification in CML

Risk stratification is crucial for CML management, particularly within HSCT. Risk assessment strategies guide treatment decisions and predict transplant outcomes. The European Society for Blood and Marrow Transplantation (EBMT) pioneered an early CML-associated allogeneic HSCT risk model, integrating donor type, disease stage, recipient age, donor–recipient gender, and diagnosis-to-transplant interval from data encompassing 3000+ patients [25]. Validated across 56,000+ transplants, this cumulative score effectively prognosticates leukemia-free survival (LFS), OS, and transplant-related mortality [26]. The lowest risk score (0) aligns with 20% transplant-related mortality and 72% 5-year OS, while the highest score (6) is associated with heightened transplant-related mortality (72%) and a modest 22% 5-year OS [25]. This EBMT score’s endorsement by the CIBMTR underscores its credibility, extending to the second allogeneic HSCT assessment for CML [27].

Notably, evaluating comorbidities during transplant gains prominence. The hematopoietic cell transplantation comorbidity index (HCT-CI), prognostic for non-relapse mortality and OS across hematologic malignancies [28], independently predicts transplant-related mortality and OS in CML HSCT, encompassing prior TKI treatment. Furthermore, elevated C-reactive protein levels during transplant predict escalated transplant-related mortality and diminished OS [29].

### 2.2. Timing of Transplant

The advent of TKIs as the primary therapy for initial CML treatment has prompted concerns about the impact of delayed HSCT on patient outcomes. Initial research, exemplified by the EBMT score [25], highlighted poorer survival when transplanting after a year from diagnosis. Amidst the contemporary landscape of widespread TKI administration prior to HSCT, a reassessment of EBMT data was conducted [30]. While the pre-TKI era showed diminished survival when transplantation exceeded a year post-diagnosis [30], this link vanished in TKI-treated patients. A systematic 2014 EBMT data reanalysis, covering 5500+ CML HSCT recipients from 2000 to 2011 during the TKI transition, disclosed similar five-year OS and progression-free survival (PFS) for TKI-treated and TKI-naive patients [31]. Interestingly, diagnosis-to-transplant time, a former TKI-naive predictor of poorer outcomes, lost significance in TKI-treated patients [31], assuaging concerns about delayed transplantation after testing second- or third-line TKIs.

While current clinical practice features TKIs as frontline CML treatment, tools exist to identify poor responders to distinct TKI generations. The analysis of newly diagnosed imatinib-treated CML patients revealed suboptimal OS associated with sustained BCR–ABL transcript levels >10% at three months, >1% at six months, and >35% Philadelphia chromosome-positive metaphases at baseline [32]. This underscores the need for intervention in non-attaining cases [32], without discouraging alternate drug trials but heightening the awareness of the potential of allogeneic HSCT for sustained remission. Emphasizing optimal pre-HSCT response timing and minimizing the gap from achievement to transplantation is pivotal. Such patients warrant early HSCT discussions and donor consideration.

### 2.3. Conditioning Regimen in CML HSCT

Early HSCT conditioning in CML primarily employs myeloablative regimens, including total body irradiation and cytotoxic agents [33]. Advances in T-cell depletion allowed for HLA-matched and mismatched unrelated donors, reducing GvHD rates with higher relapse risks [33]. To leverage the graft-versus-leukemia (GVL) effect, low-intensity conditioning and preemptive donor lymphocyte infusion (DLI) emerged. Fludarabine, low-dose busulfan, and anti-T-lymphocyte globulin in first-phase CML yielded engraftment, limited toxicity, robust GVL effects, and promising survival and disease-free outcomes, warranting prospective trials [34].

Low-intensity conditioning, particularly Fd/Bu/ATG, proved viable for early-phase CML, indicating acceptable transplant-related mortality [35]. Reduced-intensity HSCT can control chronic-phase CML, yet advanced disease demands alternative strategies considering treatment-related mortality. Patient demographic disparities underscore myeloablative vs. reduced-intensity comparisons. A retrospective analysis of the CIBMTR database showed no significant difference in OS, LFS, and non-relapse mortality between myeloablative and reduced-intensity; however, reduced-intensity conditioning had a higher risk of early relapse after allogeneic HCT (hazard ratio (HR), 1.85) and lower risk of chronic GvHD (HR, 0.77) [36]. Scarce recent data hamper assessing CML HSCT outcomes, mainly for advanced-stage patients, limiting the exploration of milder regimens (partially T-cell depleted transplant) synergizing with post-transplant TKIs to mitigate toxicity, death, and relapse risks.

### 2.4. Stem Cell Source in CML HSCT

Initially, myeloablative bone marrow-derived stem cell conditioning was standard for early CML HSCT. GVL effect recognition led to low-intensity regimens, enhancing accessibility for ineligible patients and extending safe HSCT ages [37]. While low-intensity regimens expanded the age criteria, they posed relapse risks and were unsuited for high-risk cases [38]. Myeloablative protocols remained standard where tolerable, with total body irradiation and cyclophosphamide common. Regimen choice considers patient features and institutional practice.

Peripheral blood-derived stem cells (PBSCs) replaced bone marrow, owing to quicker engraftment and donor preference. In CML and overall HSCT, PBSCs yielded elevated chronic GvHD rates, despite comparable OS and PFS [39]. Nearly half of CML HSCT patients face molecular relapse, often requiring DLI, especially after low-intensity regimens [40]. To minimize chronic GvHD and associated morbidity, impacting non-relapse mortality, cautious preference for bone marrow is suggested for first-phase CML, despite similar OS and PFS [39]. Donor-driven stem cell selection demands chronic GvHD, DLI potential, and non-relapse mortality consideration.

### 2.5. Monitoring of CML Post-HSCT

Monitoring minimal residual disease (MRD) post-HSCT is vital in CML, predicting relapse and treatment necessity. Initial qualitative polymerase chain reaction (PCR) assays for BCR–ABL1 transcripts were swiftly replaced by quantitative methods [38]. These sensitive techniques, initially for molecular relapse detection, now underpin CML monitoring. Prospective HSCT-treated CML patients revealed higher relapse risk with persistent or increasing BCR–ABL transcripts and a shorter doubling time, indicating aggressiveness [41]. Early quantitative reverse transcriptase–polymerase chain reaction (RT-PCR) post-allogeneic HSCT predicted outcomes and treatment needs based on BCR–ABL levels [30]. Correlations were observed for sibling/unrelated donor HSCT, regardless of T-cell depletion [30]. Quantifying post-HSCT BCR–ABL expression aids disease tracking and guiding decisions. Advancements and considerations are summarized in Table 1.

## 3. Indications of HSCT for CML in the Current Era

### 3.1. Cost-Effectiveness in Low-Income Countries

Imatinib’s initial price of USD 30,000 per patient/year has increased, making it financially unattainable for many [42]. Concerns persist over costly TKIs compared to imatinib [43]. Initiatives such as the Glivec International Patient Assistance Program (GIPAP) aim to bridge treatment gaps, yet challenges in disease monitoring, post-imatinib options, and HSCT access persist, hindering guideline adherence [42]. Varied resource availability leads to diverse CML management globally, echoing broader hematologic malignancy challenges.

In low-resource settings, TKIs are seen as costly. In Mexico, the first 100 days of allogeneic HSCT cost approximately USD 18,000, akin to 200 days of branded imatinib (pre-generic availability), which was extended to three years with government-backed generic imatinib [42]. The Czech Republic findings align, showing that imatinib was 25% pricier than reduced-intensity conditioning HSCT during the initial two years [40]. Similar trends emerge in China, Eastern Europe, and Latin America, favoring HSCT due to lower CML treatment costs [42,44].

A crucial aspect to consider is the cost comparison between the first year of TKI treatment and HSCT. While HSCT is undeniably cost-efficient in middle- and low-income countries, the cost comparison is equally significant in developed countries due to its impact on the healthcare system. For instance, a study conducted in Japan and the United States of America (USA) highlights this issue [45]. In the USA, starting treatment with imatinib first resulted in 7.34 quality-adjusted life years (QALYs) at a cost of USD 1,022,148, and in Japan, it cost JPY 32,526,785. Comparatively, dasatinib first yielded 7.68 QALYs at a cost of USD 1,236,052 (or JPY 51,506,254), nilotinib first achieved 7.64 QALYs at a cost of USD 1,245,667 (or JPY 39,635,598), and physician’s choice provided 7.55 QALYs at a cost of USD 1,167,818 (or JPY 41,187,740) in the USA [45]. In Japan, these strategies incurred significantly higher costs. None of these approaches met the willingness-to-pay threshold [45]. Importantly, imatinib stood out as the most cost-effective option, even when considering the possibility of discontinuing TKIs in the future.

Cost-effectiveness strongly favors allogeneic HSCT over lifelong TKIs, particularly in developing countries. Developed nations also see HSCT as cost-effective, notably in youth [38]. Strategies such as reduced-intensity conditioning, outpatient HSCT, and peripheral blood stem cells enhance HSCT affordability and accessibility.

### 3.2. Children and Young Adults

Pediatric TKI use is comparable in safety and efficacy to adults, yet HSCT also exhibits effectiveness. Myeloablative HSCT in early CML patients aged <18 and 18–29 showed 75% 5-year OS and 59% LFS [46]. Comparative HSCT–imatinib analysis in pediatric CML indicated similar 84% (HSCT) and 87% (imatinib) two-year OS and relapse rates [47]. Allogeneic HSCT in pediatric CML showed 97.4% 5-year OS and 79.8% EFS, without mortality at 100 days/1-year post-HSCT [48]. Allogeneic HSCT’s chronic-phase CML effectiveness, for TKI failure, reported 84% (adults) and 91% (pediatrics) pooled OS in quantitative synthesis [49].

A survey of pediatric oncologists and HSCT physicians revealed that HSCT preference for pediatric CML is a minority, yet they were interested in a DMR-driven discontinuation trial [50]. Lifelong TKI treatment demands, especially from childhood to young adulthood, could lead to growth issues, potentially favoring HSCT. The suitability of HSCT as a TKI alternative, particularly for those diagnosed young, should consider the advantages of TFR over HSCT and the drawbacks of total body irradiation (TBI) [51]. While TBI-associated complications pose concerns, HSCT’s promising long-term OS in chronic CML may appeal to young patients. Notably, opting for HSCT or HSCT due to imatinib failure resulted in comparable survival rates [52].

### 3.3. Aiming for TFR in CML

TFR in CML holds promise, with approximately 50% achieving DMR-sustaining TFR [53]. Imatinib discontinuation in durable molecular response 4.5 (MR4.5) patients led to relapse rates of 17% (continuation) and 67% (discontinuation), which responded to imatinib reintroduction [54]. CMR patients discontinuing imatinib showed 61% relapse within a median of 17 months but responded to reinitiation, suggesting sustained CMR [55]. Some non-responders might be considered operationally cured, given prolonged leukemia-free survival [56]. The likelihood of an operational cure in second-generation TKI-treated patients is plausible. While large studies have explored second-generation TKI discontinuation, a small study has demonstrated its feasibility [57].

TKIs significantly impact CML, but they are not curative, necessitating indefinite therapy. For discontinuation trials aiming at TFR, specific criteria encompass 3-year TKI duration, 2-year sustained DMR, and MR4 [58]. However, TFR achievement in newly diagnosed patients is limited (20–30%) [58]. Starting CML patients should discuss lifelong TKI likelihood for informed decisions. Evidence indicates that most patients require indefinite TKI use, making HSCT a potential cure-seeking option. Such discourse prepares for possible HSCT post-TKI failure while considering lifelong TKI challenges, empowering informed decisions.

### 3.4. Intolerance and Resistance to TKIs in Chronic-Phase CML

Three TKIs—imatinib, dasatinib, and nilotinib—are approved for first-line chronic-phase CML, focusing on deep responses [59]. Imatinib is common, yielding 70% complete cytogenetic response (CCyR) at 12 months, but 40% encounter imatinib failure by five years [60,61,62]. Second-line TKIs such as dasatinib, nilotinib, or bosutinib can provide long-term responses for half of imatinib-intolerant/failure cases. While imatinib achieves CCyR, dasatinib and nilotinib offer superior responses, with similar adherence [63,64]. TKI discontinuation due to side effects or resistance is seen, with primary/secondary resistance linked to kinase domain mutations. HSCT benefits approximately 20% of TKI-ineffective patients [65].

Early response assessment gains importance in predicting CML outcomes [56]. Monitoring BCR–ABL1 levels after three months of second-line therapy can help identify HSCT candidates. Approximately 10–15% of patients in the chronic phase will not achieve durable remission, making HSCT viable, especially for multiple TKI-resistant/intolerant cases [66]. Poor OS is linked to BCR–ABL1/ABL1 >10% (imatinib) and >2% (dasatinib/nilotinib) at three months [67]. Early second-line response predicts long-term outcomes, classifying risk based on the 3-month transcript ratio. Monitoring aids non-responder identification, initiating stem cell donor search within three months.

Even in TKI-resistant/intolerant CML, HSCT thrives. A meta-analysis showed HSCT efficacy for chronic-phase CML in adults with a pooled 84% OS, 66% DFS, and a pediatric OS of 91% [49]. Evolving CML failure management incorporates HSCT early [68]. Discussion on TKI response, treatment switches, and HSCT benefits is essential. Strategic frontline treatment and prompt intensification prevent progression and enhance outcomes.

### 3.5. Blast Crisis

Blast crisis in CML arises from persistent BCR–ABL1 activity, inducing genomic instability and karyotypic anomalies [38]. The WHO’s classification for blast phase is characterized by one of the following criteria: myeloid blasts comprising 20% or more of the total cells in the blood or bone marrow; the presence of extramedullary proliferation of blasts; or the existence of increased lymphoblasts in the peripheral blood or bone marrow [6]. Distinguishing the minimal blast proliferation for the definition of blast phase is crucial, as the WHO and other references differ (30% cutoff) [69]. Blast-phase CML is rare, representing 2.2% of CML cases with blast crisis [70]. Approximately 3.1% of chronic CML patients progress to the blast phase despite TKI treatment. TKIs yield CCyR in 7% to 37% of blast-phase CML cases, particularly with second-generation TKIs [71].

Despite advances in treating chronic-phase CML, managing blast crisis remains challenging, and employing chemotherapy regimens similar to those used in acute leukemia may enhance response rates. The goal is a second chronic phase followed by HSCT for eligible patients. Even with TKIs, outcomes are modestly improved, yet survival remains limited. The response to TKIs and HSCT involves ABL1 kinase gene mutations, cytogenetic abnormalities, and blast-phase involvement [6].

Ideally, patients nearing the blast phase should initiate TKIs, followed by HSCT consideration. Achieving a second chronic phase before HSCT improves outcomes, given the bleak survival rates (<10%) in frank blast-crisis HSCT [72,73]. Novel TKIs or combined chemotherapy can restore the second chronic phase before transplantation, using TKIs, chemotherapy, or their combination [38]. HSCT in the second chronic phase shows enhanced overall and LFS rates (36% and 27%) based on CIBMTR data (1999–2004) [73]. An EBMT study reported a 2-year survival estimate of 47% for second/subsequent chronic-phase transplants [74]. Although registry studies lack phase-transition proportions, some patients did not benefit from TKIs. Recent data show a better prognosis in the imatinib-induced second chronic phase but with limited follow-up.

For those not achieving CCyR, allogeneic HSCT is the sole potential avenue for long-term survival. While fully myeloablative conditioning is preferred if tolerated, registry studies indicate comparable survival with reduced-intensity conditioning [35,75]. Exploring newer TKIs post-transplantation is also viable [35,75]. In the myeloproliferative neoplasm blast phase, allogeneic HSCT is the primary route to long-term remission, yielding a 3-year OS of 36% in 663 patients; favorable outcomes are linked to recent allogeneic HSCT, Karnofsky performance score ≥90, and complete response at transplantation [76]. Notably, a study comparing allogeneic HSCT outcomes in blast-crisis CML between haploidentical and matched-related donors showed similar 3-year OS (60.0% vs. 55.3%) and RFS rates (51.1% vs. 47.8%), suggesting that haploidentical donors are viable for selected patients [77].

The WHO’s 2022 blast CML definition incorporates myeloid blasts exceeding 20% and factors such as extramedullary infiltration or increased lymphoblasts, distinguishing myeloid and lymphoid phases [6]. Strong recommendations await endorsement due to limited large-scale studies, hindering conclusive results. Most studies also lack analysis based on these types. Notably, lymphoid blast CML presents better survival rates (5-year rates: 15% vs. 30%) than myeloid blast CML, with potential future targeted therapies indicated by notable genetic differences [78].

Observational studies indicate that 49% of lymphoid blast-phase patients receive combined TKI, chemotherapy, and HSCT treatment, while chemotherapy and dasatinib are favored for lymphoid over myeloid blast CML [79]. Adjunctive approaches enhance CCyR rates but are associated with shorter survival than TKIs alone [71]. Ongoing research explores chemo-TKI combinations, including asciminib (first-in-class specific allosteric inhibitor) and B-cell lymphoma inhibitors, for the lymphoid blast phase [79]. The myeloid blast phase (70% cases) prompts TKI use based on prior therapy or mutations, possibly adding chemotherapy such as dasatinib or ponatinib [24]. Following the second chronic phase, immediate transplantation is advised, and in the UK, FLAG-IDA chemotherapy with TKIs is standard; BSH guidelines recommend HSCT irrespective of the initial response, suggesting CNS prophylaxis for the myeloid and lymphoid phenotypic blast phase [24,80].

While TKIs effectively treat chronic-phase CML and reduce blast-crisis instances, their efficacy in blast crisis is limited and transient, warranting novel strategies and re-evaluation of blast crisis and treatment failure [81]. Post-transplant mortality primarily stems from transplant-related causes, with a 1-year mortality rate of 46% during blast-crisis transplantation, improving to 33% if a second chronic phase is attained prior to transplantation [73]. Despite transplant-related risks, allogeneic HSCT remains the vital option for eligible blast-crisis CML patients, representing the sole avenue for long-term survival, recognized as the “gold standard” for advanced disease stages. While some accelerated-phase CML patients achieve long-term TKI responses, HSCT is unequivocally recommended as the curative approach for blast-crisis CML, preferably after the initial TKI response [68]. For relapsing and refractory cases, phase I and II trials with chemotherapy, immunotherapy, or monoclonal antibodies are advisable [71].

### 3.6. Advanced Accelerated-Phase CML

ELN defines accelerated CML as the presence of peripheral blood or bone marrow blasts comprising 15–29% of the cell count, or a combination of blasts and promyelocytes exceeding 30%, with blasts accounting for less than 30% alone. Additionally, it is characterized by an elevated peripheral blood basophils count of 20% or higher, platelet levels below 100 × 10^9^/L not attributed to treatment, and the appearance of additional genetic abnormalities during treatment, indicating clonal evolution. It is worth noting that the latest WHO update has combined the accelerated phase and chronic phase into a single category. This phase covers a diverse patient cohort [82]. HSCT provides optimal long-term survival prospects for those approaching blast-crisis transformation, whereas TKI-based approaches can be suitable for early-stage transitions from chronic to accelerated phases [38].

Heterogeneous research on TKI efficacy in the accelerated phase highlights dependence on proximity to blast crisis. Some studies demonstrate favorable 3-year OS rates with imatinib or second-generation TKIs in early accelerated phases, but this strategy suits select patients [83]. Other studies indicate a better prognosis for hematologically defined accelerated phases over those with additional cytogenetic abnormalities [84]. Approximately 20–30% of patients progressing to this phase after the initial imatinib treatment achieve CCyR with dasatinib or nilotinib [85,86]. The cumulative best cytogenetic response to imatinib was 21%, with seven-year EFS and OS rates of 15% and 45%, respectively; higher imatinib doses correlate with improved responses and survival [87].

The efficacy of HSCT surpasses that of TKIs, particularly in the late accelerated phase. In a Beijing study, HSCT showed no benefit for low-risk patients (CML duration ≤12 months, hemoglobin >100 g/L, and peripheral blood blasts ≤5%) over imatinib alone; however, HSCT salvaged patients with two or more high-risk factors and poor outcomes [88]. Among adults with accelerated-phase CML, allogeneic HSCT yielded significantly better outcomes than second-line TKIs (nilotinib and dasatinib), with 5-year OS rates of 86.4% vs. 42.9%, EFS rates of 76.1% vs. 14.3%, and PFS rates of 78.1% vs. 28.6%, respectively [89]. In a recent CIBMTR retrospective study, 185 patients transplanted in the accelerated phase showed 3-year OS and LFS rates of 43% and 37%, respectively [73].

TKI therapy stands as a viable option for patients in the early accelerated phase who are new to TKI treatment. Monitoring the molecular response at three and six months can identify those with favorable long-term outcomes. Low-risk patients achieving an optimal response could sustain TKI treatment alone, although this criterion is established primarily for chronic-phase patients [82]. Commencing donor searches and vigilant monitoring are advisable in these cases.

### 3.7. T315I Mutation in CML

The T315I mutation in CML leads to TKI resistance, yet the third-generation TKI ponatinib offers a durable response [90]. Ponatinib’s effectiveness in chronic-phase CML with the T315I mutation makes it a preferred initial choice, although vascular events warrant caution in high-risk cases. HSCT presents a potentially curative avenue for CML patients harboring BCR–ABL T315I mutations. Encouraging results have been obtained in the chronic phase, compared with moderate response rates in the accelerated phase and bleak outcomes in the blast phase [91].

In assessing ponatinib and HSCT efficacy for T315I-positive CML, a retrospective study compared chronic-phase CML and Philadelphia-positive acute lymphoblastic leukemia cases. Ponatinib displayed superior 24-month and 48-month OS rates compared to HSCT for T315I-positive chronic-phase CML, offering a valuable alternative treatment option. Nevertheless, in blast-crisis CML and Philadelphia-positive acute lymphoblastic leukemia with the T315I mutation, ponatinib’s OS was shorter than that of HSCT, underscoring the continued significance of HSCT in advanced cases [92].

Hence, for chronic-phase CML with the T315I mutation, ponatinib could serve as an initial treatment option before considering HSCT. Nevertheless, early HSCT upon T315I detection may enhance survival prospects, especially given the bleak outlook for blast-phase patients with this mutation [91]. In advanced CML stages, allogeneic HSCT stands as the sole avenue for prolonged survival.

### 3.8. Concurrent Myelodysplastic Syndromes with CML

Limited data exist regarding the optimal treatment strategy for patients with concurrent lymphoid and myeloid conditions, such as co-occurring CML and myelodysplastic syndromes (MDSs), particularly when prior chemotherapy or radiotherapy is absent. Case reports have documented such instances [93]. In one clinical trial, eight patients underwent allogeneic HSCT for simultaneous myeloid and lymphoid malignancies, two of whom had active primary neoplasms during transplantation. In both cases, HSCT led to complete remission for both diseases [94]. Another case series involving two patients with active myeloid and lymphatic malignancies at the time of HSCT revealed that HSCT can induce prolonged remission for both malignancies, even with active disease during transplantation, although remission attainment time can differ across malignancies [95].

Approximately 5% of patients with CCyR exhibit Philadelphia chromosome-negative clonal changes, which usually do not significantly affect the course of CML. However, in rare instances, these changes may involve myelodysplastic features, warranting HSCT consideration. Allogeneic HSCT provides curative potential for concurrent MDS and lymphoid malignancies, with remission observed in previously untreated and treated patients, although the high non-relapse mortality in the latter calls for improved transplant strategies [96]. Patients with two active neoplasms during HSCT seem to have comparable prognoses to those with a single myeloid disorder. Factors such as prior cytotoxic therapy, conditioning regimen, and chronic graft-versus-host disease influence outcomes. These findings highlight the need for more research on prognostic indicators and optimal HSCT approaches for this patient group. Despite limited studies, early HSCT offers a potential cure for concurrent leukemias, although guidelines remain unclear, and survival outcomes are poor. Table 2 summarizes all indications for HSCT in CML.

## 4. Conclusions and Further Considerations

In conclusion, HSCT remains a valuable treatment option for CML patients, particularly in high-risk cases, those with unsatisfactory responses to TKIs, and those in advanced disease stages. Risk stratification models like the EBMT score and the HCT-CI are valuable tools for predicting transplant outcomes. The timing of HSCT after TKI therapy, the choice of conditioning regimen and stem cell source, and close monitoring for minimal residual disease post-HSCT are crucial considerations. Furthermore, HSCT demonstrates efficacy in TKI-resistant or intolerant CML patients and is vital for managing blast crises and advanced accelerated-phase CML. While HSCT’s cost-effectiveness in low-income countries and its potential benefits for children and young adults are important, the decision to pursue HSCT can vary based on individual circumstances. Additionally, gene mutations in CML patients present complex treatment challenges, and HSCT may be considered, especially when optimal responses to TKIs are unlikely. The considerations of lifelong TKI therapy, pregnancy possibilities, and the social impact of being a cured patient should guide the decision-making process. Table 3 summarizes some of the clinical practice points and future research ideas.

## Figures and Tables

**Table 1 cancers-16-00754-t001:** Summary of the Advancements and Considerations in HSCT for CML.

Topic	Subsection	Key Points
Risk stratification in CML	EBMT risk assessment model	− Cumulative score based on donor type, disease stage, recipient age, donor and recipient sex, time to transplantation
	CIBMTR validation	− Validated as a predictor of leukemia-free survival, overall survival, and transplant-related mortality
	HCT-CI evaluation	− HCT-CI predicts non-relapse mortality and overall survival in CML patients undergoing HSCT
	CRP Levels	− Elevated C-reactive protein at transplant predicts increased mortality and decreased overall survival
Timing of Transplant	Impact of TKIs on transplant outcomes	− Postponing transplantation after TKI treatment does not negatively impact transplant outcomes
	Identifying poor TKI responders	− Patients with poor response milestones may benefit from early HSCT
Conditioning Regimen in HSCT	Myeloablative regimens	− Early HSCTs used myeloablative conditioning with total body irradiation and cyclophosphamide
	Low-intensity conditioning	− Low-intensity conditioning with fludarabine, busulfan, and ATG shows promise
	Reduced intensity for high-risk patients	− Reduced-intensity protocols are suitable for good-risk patients and those in the first or second chronic phase
	PBSC vs. bone marrow	− PBSCs provide faster engraftment but have higher rates of chronic GvHD
Stem Cell Source Choice in HSCT	Historical stem cell source choice	− Myeloablative conditioning used bone marrow-derived stem cells
	Low-intensity impact on stem cell source choice	− Reduced-intensity regimens led to increased use of PBSCs
	Caution with PBSCs in CML HSCT	− PBSCs associated with higher chronic GvHD risk and non-relapse mortality
Monitoring of CML post-HSCT	Importance of MRD monitoring	− Quantifying BCR–ABL transcripts helps predict relapse and guide treatment decisions
	BCR–ABL and relapse probability	− Persistent or increasing BCR–ABL transcripts indicate higher relapse rates
	Early RT-PCR for outcome prediction	− Early RT-PCR testing effectively predicts long-term outcomes after HSCT

Caption: This table summarizes the key points in HSCT for CML, including risk stratification models, the timing of transplantation, conditioning regimens, stem cell source choices, and the importance of monitoring minimal residual disease (MRD) post-HSCT. Abbreviations: HSCT, hematopoietic stem cell transplantation; CML, chronic myeloid leukemia; EBMT, European Society for Blood and Marrow Transplantation; CIBMTR, Center for International Blood and Marrow Transplant Research; HCT-CI, hematopoietic cell transplantation comorbidity index; TKIs, tyrosine kinase inhibitors; GvHD, graft-versus-host disease; DLI, donor lymphocyte infusion; MRD, minimal residual disease; PCR, polymerase chain reaction; RT-PCR, reverse transcriptase–polymerase chain reaction; PBSC, peripheral blood-derived stem cell.

**Table 2 cancers-16-00754-t002:** Indications of HSCT for CML in the Current Era.

Indication	Summary
Cost-effectiveness in low-income countries	Financial barriers hinder access to TKIs in developing countries, making HSCT a more feasible and cost-effective option.
Children and young adults	HSCT demonstrates favorable outcomes and may be preferred over lifelong TKI therapy for pediatric and young adult patients.
Aiming for TFR in CML	Achieving treatment-free remission is a goal; however, HSCT remains the only curative therapy for long-term remission.
Intolerance and resistance of TKIs	HSCT is recommended for patients resistant or intolerant to TKIs, providing better long-term survival opportunities.
Blast crisis	HSCT is vital for patients in blast crisis, achieving long-term remission and improved survival rates.
Advanced accelerated-phase CML	HSCT offers better outcomes than TKIs, particularly in late accelerated-phase cases.
T315I mutation in CML	Ponatinib is effective for T315I-positive chronic-phase CML, but HSCT should be considered for Ponatinib resistance and advanced stages of CML.
Concurrent myelodysplastic syndromes	HSCT can lead to complete remission of both myeloid and lymphoid malignancies, providing a potential, curative option.

Caption: Summary of indications for HSCT in CML based on different clinical scenarios and disease stages. HSCT offers curative potential, better outcomes, and cost-effectiveness in specific patient populations. Abbreviations: TKIs, tyrosine kinase inhibitors; HSCT, hematopoietic stem cell transplantation; TFR, treatment-free remission; CCyR, complete cytogenetic response; OS, overall survival; LFS, leukemia-free survival; EFS, event-free survival; CMR, complete molecular remission; MDS, myelodysplastic syndrome.

**Table 3 cancers-16-00754-t003:** Clinical practice points and future research suggestions for HSCT in CML.

Clinical Practice Points	Future Research Suggestions
− Consider HSCT in middle and low-income nations for cost-effectiveness.	− Study the effectiveness of a partially T-cell-depleted conditioning regimen in chronic CML through clinical trials.
− Optimize HSCT cost-effectiveness using reduced-intensity conditioning, outpatient management, peripheral blood stem cells, and improved accessibility.	− Explore novel treatment approaches or combinations for blast-crisis CML due to poor prognosis with HSCT and TKIs.
− Initiate discussions about HSCT with CML patients, considering factors like TFR, social impact, lifelong TKI therapy, and future pregnancies.	− Categorize blast-crisis CML patients based on phenotype (myeloid, lymphoid, and mixed) for subsequent investigations.
− Recommend HSCT after two TKI failures in chronic-phase CML due to minimal likelihood of TFR.	− Retrospectively identify and compare outcomes between phenotype groups (myeloid, lymphoid, and mixed) in blast-crisis CML trials.
− Achieve second chronic phase in CML blast-phase patients before HSCT.	− Collect additional data on CML patients with additional gene mutations.

Caption: Recommendations and research agenda for HSCT in CML. Abbreviations: TKIs, tyrosine kinase inhibitors; HSCT, hematopoietic stem cell transplantation; TFR, treatment-free remission; CML, chronic myeloid leukemia.

## Data Availability

Data are contained within the article.

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
