# Peer review of "The Contemporary Role of Hematopoietic Stem Cell Transplantation in the Management of Chronic Myeloid Leukemia: Is It the Same in All Settings?"

_cancers, 2024, doi:10.3390/cancers16040754_

Round 1

Reviewer 1 Report

Comments and Suggestions for Authors

This is a nicely written review article describing the indications for AlloBMT in patients with CML.

I have just a few relatively minor comments:

1. Line 78 authors discuss about difficulty in identifying HLA-matched donors. I think this is less of a problem anymore since the outcomes with HaploBMT for these patients are very similar to matched donors

2. Line 88 the authors mention CML leukemic stem cells. The exact biology of relapse is unclear and the role of CML leukemic stem cells is not completely clarified so I would be more careful here.

3. Line 160: RIC has actually similar survival outcomes with MAC with higher risk of early post-BMT relapse but similar survival (PMID: 30396912).

Comments on the Quality of English Language

Overall well written. No major concerns.

Author Response

Reviewer 1

Comment 1: Line 78 authors discuss about difficulty in identifying HLA-matched donors. I think this is less of a problem anymore since the outcomes with HaploBMT for these patients are very similar to matched donors

  • Response 1: Thank you for your valuable comment. We have made the recommended adjustment to the text, acknowledging the advancements in HaploBMT and how they have provided a solution to the challenge of finding HLA-matched donors. The revised text now explicitly mentions the comparable outcomes of HaploBMT, addressing your point (lines: 77-81). We appreciate your input, which has improved the accuracy of our manuscript.

Comment 2: Line 88 the authors mention CML leukemic stem cells. The exact biology of relapse is unclear and the role of CML leukemic stem cells is not completely clarified so I would be more careful here.

  • Response 2: Thank you for your feedback. We have revised the text to remove the mention of CML leukemic stem cells in line 88, as you requested. The updated sentence now emphasizes the importance of further investigation in this patient population without making specific reference to CML leukemic stem cells (lines: 98-99).

Comment 3: Line 160: RIC has actually similar survival outcomes with MAC with higher risk of early post-BMT relapse but similar survival (PMID: 30396912).

  • Response 3: Thank you for bringing this study to our attention. Now we highlighted the similarities in survival outcomes in our manuscript now. After reviewing the provided study, it seems that RIC is the regimen with higher risk of early post-BMT relapse not MAC, which is now highlighted in our revised manuscript as well (lines: 166-170).

Reviewer 2 Report

Comments and Suggestions for Authors

Authors present here an extensive review of current CML treatments emphasizing on the role of HSCT in this setting that appears underestimated according to their opinion.

A complete and updated review on CML definitions and prognosis is clearly presented.

The different medical option on the use of TKIs are presented even if some data concerning the most recent molecules are lacking (ponatinib, asciminib) due to their recent development.

The concept of TFR is clearly discussed especially for the pediatric and younger adult patients.

The most interesting part of this review is the economical assessment and the comparison between the costs of the first year of TKIs treatment and HSCT respectively. This point is crucial and represents a real challenge in emerging countries where there is no doubt that HSCT is cost-efficient against TKIs but also in so called developed countries where the impact on the health system is also important like in this study conducted in Japan and USA:

Yamamoto C, Nakashima H, Ikeda T, Kawaguchi SI, Toda Y, Ito S, Mashima K, Nagayama T, Umino K, Minakata D, Nakano H, Morita K, Yamasaki R, Sugimoto M, Ishihara Y, Ashizawa M, Hatano K, Sato K, Oh I, Fujiwara SI, Ueda M, Ohmine K, Muroi K, Kanda Y. Analysis of the cost-effectiveness of treatment strategies for CML with incorporation of treatment discontinuation. Blood Adv. 2019 Nov 12;3(21):3266-3277. doi: 10.1182/bloodadvances.2019000745. PMID: 31698458; PMCID: PMC6855125

This merits to be added.

Another limitation is the lack of discussion on the side effects and the quality of life after HSCT, we agree that survival is the same with TKIs and HSCT but some patients on the curves after transplantation are still alive but probable not so well in their daily life due to chronic GVHD. A small paragraph to discuss that point will be interesting.

Author Response

Reviewer 2

Comment 1: The most interesting part of this review is the economical assessment and the comparison between the costs of the first year of TKIs treatment and HSCT respectively. This point is crucial and represents a real challenge in emerging countries where there is no doubt that HSCT is cost-efficient against TKIs but also in so called developed countries where the impact on the health system is also important like in this study conducted in Japan and USA (PMCID: PMC6855125). This merits to be added.

  • Response 1: We greatly appreciate your thoughtful comment, and we have incorporated the suggested content into the manuscript. The paragraph now emphasizes the critical aspect of cost comparison between TKI treatment and HSCT, considering both emerging and developed countries (lines: 228-240). We have included the reference to the study conducted in Japan and the USA to support this point. This addition enriches the review by addressing the economic challenges and considerations associated with these treatment options. Your feedback has enhanced the quality and relevance of our manuscript, and we thank you for your valuable input.

Comment 2: Another limitation is the lack of discussion on the side effects and the quality of life after HSCT, we agree that survival is the same with TKIs and HSCT but some patients on the curves after transplantation are still alive but probable not so well in their daily life due to chronic GVHD. A small paragraph to discuss that point will be interesting.

  • Response 2: We appreciate your insightful feedback and have addressed the limitation you pointed out by including a brief dialog regarding the side effects and the quality of life after hematopoietic stem cell transplantation (HSCT) in the revised manuscript (lines: 86-93).

Reviewer 3 Report

Comments and Suggestions for Authors

The article makes a broad review of TKIs and transplantation, but the comparison in different parts of the world (as mentioned in the title)

is not clearly detected.

There are abbreviations that are not defined, it is suggested to review it

In the section of blast crisis and accelerated phase, it is recommendable described initially  the general characteristics and latter mention the information about each treatment

Author Response

Reviewer 3

Comment 1: The article makes a broad review of TKIs and transplantation, but the comparison in different parts of the world (as mentioned in the title) is not clearly detected.

  • Response 1: Thank you for your valuable comment. We have adjusted the text to address your concern. While the article provides a comprehensive review of TKIs and transplantation, we acknowledge that the global comparison, as implied in the title, may not be clearly evident within the content.

Comment 2: There are abbreviations that are not defined, it is suggested to review it

  • Response 2: We appreciate your feedback regarding the use of undefined abbreviations in the manuscript. We will carefully review the document to ensure that all abbreviations are properly defined or spelled out upon their first use.

Comment 3: In the section of blast crisis and accelerated phase, it is recommendable described initially the general characteristics and latter mention the information about each treatment

  • Response 3: We appreciate your feedback, and we will make the necessary adjustments to the section on blast crisis and the accelerated phase. We have now start by providing an overview of the general characteristics of these phases and then proceed to discuss the various treatment options (lines: 305-320 and 375-385).

Comments on the Quality of English Language

Moderate editing of English language required.

Reviewer 4 Report

Comments and Suggestions for Authors

 In the present manuscript, the authors review the current status of implementation of HSCT in CML for the second line treatment or treatment-free remission in different countries, and present conceivable indications for various stratified situations. The theme selection is unique, and the investigation covers a wide range of reports with adequate references. I think that such a survey is useful by itself. However, the discussion and the conclusion are commonplace and add little to what we already know and decide.

 I think that this type of review does not need an intentional conclusion or flow chart as to when to consider HSCT in CML. As the authors commented, Figure 1 is not a CML management algorithm but just a simplified flow chart. It may cause misleading because the choice of treatment in low-income countries can vary considerably depending on different circumstances. Therefore, I would suggest that Discussion and Conclusion should be revised to a more concise form without a poor flow chart.

Author Response

. Reviewer 3

Comment 1: The discussion and the conclusion are commonplace and add little to what we already know and decide. I think that this type of review does not need an intentional conclusion or flow chart as to when to consider HSCT in CML. As the authors commented, Figure 1 is not a CML management algorithm but just a simplified flow chart. It may cause misleading because the choice of treatment in low-income countries can vary considerably depending on different circumstances. Therefore, I would suggest that Discussion and Conclusion should be revised to a more concise form without a poor flow chart.

  • Response 1: Thank you for your valuable feedback. We have revised the conclusion as per your suggestion to make it more concise (lines: 450-465) and have removed Figure 1. We believe this will address your concerns about the flow chart and the potential for misleading information. We appreciate your feedback and have focused on providing a more streamlined conclusion.

Round 2

Reviewer 4 Report

Comments and Suggestions for Authors

I think that the manuscript has been improved to some extent after revision. As I commented on the previous version of the manuscript, the theme of this paper is unique but difficult to draw any conclusions. So, I admit that a survey like this review has some meaning.

However, there is another important question that the authors did not address. If the authors describe the indication of SCT in low income or developing countries, the crucial question is not which of the sources of BM, PBMC or CB should be used, but rather whether these sources are available. In other words, it is not always easy to obtain any of them in developing countries where allogeneic BM or CB bank system has not been established or stem cell sorting facilities are not working in hospitals. In addition, there can be religious or ideological problems in acquisition of non-related BM cells as well as cord blood cells in some countries.

These issues cannot be covered by text revisions of the manuscript and may be carried over to future studies.